# DIPPER: Direct Preference Optimization for Primitive-Enabled Hierarchical Reinforcement Learning

## Abstract

Hierarchical reinforcement learning (HRL) is an elegant framework for learning efficient control policies to perform complex robotic tasks, especially in sparse reward settings. However, concurrently learning policies at multiple hierarchical levels often suffers from training instability due to non-stationary behavior of lower-level primitives. In this work, we introduce DIPPER, an efficient hierarchical framework that leverages Direct Preference Optimization (DPO) to mitigate non-stationarity at the higher level, while using reinforcement learning to train the corresponding primitives at the lower level. We observe that directly applying DPO to the higher level in HRL is ineffective and leads to infeasible subgoal generation issues. To address this, we develop a novel, principled framework based on lower-level primitive regularization of upper-level policy learning. We provide a theoretical justification for the proposed framework utilizing bi-level optimization. The application of DPO also necessitates the development of a novel reference policy formulation for feasible subgoal generation. To validate our approach, we conduct extensive experimental analyses on a variety of challenging, sparse-reward robotic navigation and manipulation tasks. Our results demonstrate that DIPPER shows impressive performance and demonstrates an improvement of up to 40% over the baselines in complex sparse robotic control tasks.

## 1 Introduction

The success of deep reinforcement learning (RL) is impeded in sparse reward scenarios due to limitations like ineffective exploration and long-term credit assignment (Gu et al., 2016; Levine et al., 2015; Nair et al., 2018). To overcome these issues, Hierarchical reinforcement learning (Sutton et al., 1999; Harb et al., 2018) is an elegant framework which promises the benefits of temporal abstraction and improved exploration (Nachum et al., 2019) . In the goal-conditioned hierarchical RL setting (Dayan and Hinton, 1992; Vezhnevets et al., 2017) that we consider in this paper, the higher-level policy provides subgoals to a lower-level policy, which in turn tries to achieve those subgoals by executing primitive actions. Off-policy hierarchical reinforcement learning (HRL) approaches (Levy et al., 2018; Nachum et al., 2018) face significant limitations, including: **Limitation L1:** non-stationarity due to evolving lower-level primitive policy, and **Limitation L2:** infeasible subgoal generation by higher-level policy(Chane-Sane et al., 2021). When the higher and lower level policies are trained concurrently in HRL, due to continuously changing and sub-optimal lower level policy, the higher level reward function and transition model become non-stationary. This phenomenon is called non-stationarity in HRL. Further, the higher level policy may generate subgoals that are too hard for the lower primitive to achieve, a phenomenon referred to as infeasible subgoal generation.

A recent work by Singh et al. (Singh et al., 2024) attempts to mitigate Limitation L1 by leveraging preference learning ideas from reinforcement learning from human feedback (RLHF) (Christiano et al., 2017; Lee et al., 2021). Specifically, their key idea is to utilize preference-based human feedback to learn a reward function for the higher level, thereby avoiding reliance on the lower-level policy for higher-level reward computation. While shown to be effective, this approach introduces an additional bottleneck: first, the higher-level reward function must be learned from preference feedback, and then reinforcement learning is employed to optimize this reward to learn the higher-level optimal policy. Moreover, the optimal policy learned from the preference-feedback-based reward

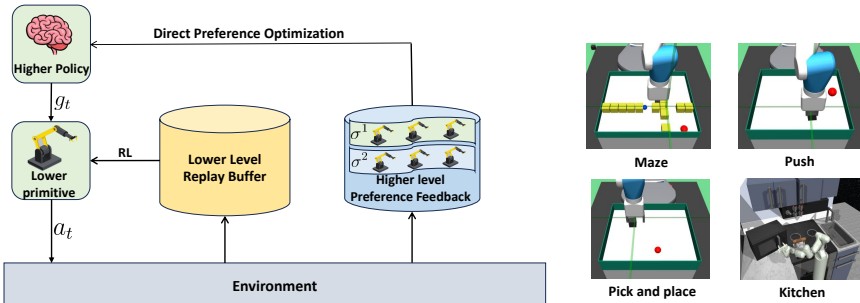

Figure 1: **DIPPER overview (left):** The higher-level policy predicts subgoals $g_t$ for the lower-level policy, which executes primitive actions $a_t$ on the environment. The lower-level policy's replay buffer is populated by environment interactions, and is optimized using RL. Further, using the elicited preference dataset, direct preference optimization is used to learn the higher-level policy. **Training environments (right):** $(i)$ maze navigation, $(ii)$ pick and place, $(iii)$ push, and $(iv)$ franka kitchen environment.

may still suffer from infeasible subgoal generation, failing to address Limitation L2. Hence, we pose the following question: *Is there an efficient hierarchical approach for solving robotic control tasks using human preference data that simultaneously addresses the issues of non-stationarity and infeasible subgoal generation in hierarchical reinforcement learning (HRL)?*

In this work, we affirmatively answer the above question by proposing DIPPER: **DI**rect **P**reference Optimization to Accelerate **P**rimitive-**E**nabled Hierarchical **R**einforcement Learning. DIPPER employs Direct Preference Optimization (DPO) (Rafailov et al., 2024b) to learn the higher-level policy and RL to learn the lower-level policy. The key insight is that by leveraging DPO to learn the higher-level policy using preference data, DIPPER decouples higher-level policy from the non-stationary lower-level primitives, thereby mitigating non-stationarity (addressing Limitation L1) in HRL.

Further, to address Limitation L2, we regularize the higher-level policy to predict feasible subgoals to the lower-level policy. We provide a theoretical justification for this regularization via a bi-level optimization formulation of HRL. The regularization term ensures that we maximize higher-level rewards while constraining the lower-level primitives to remain close to optimal. Using this bi-level formulation, we also derive a novel reference policy for DPO that regularizes the higher-level policy to generate feasible subgoals: which we call primitive regularization.

To summarize, the main contributions of this work are as follows.

**1. Novel hierarchical approach (DIPPER):** We introduce DIPPER, a new hierarchical framework for solving complex robotic control tasks using direct preference optimization (Section 4).

**2. Mitigation of non-stationarity in HRL:** We show that DIPPER is able to mitigate the effect of non-stationarity inherent in off-policy HRL in a variety of scenarios (Section 5).

**3. Mitigation of infeasible subgoal generation in HRL:** Utilizing our bi-level optimization formulation, we derive a primitive-enabled reference policy that regularizes the higher-level policy to generate feasible subgoals (Section 4.1.2).

**4. Empirical success in complex tasks:** We experimentally demonstrate that DIPPER demonstrates an improvement of upto 40% over the baselines in most of the task environments, outperforming existing baselines that typically fail to make significant progress (Section 5).

## 2 RELATED WORKS

**Hierarchical Reinforcement Learning.** HRL provides an elegant framework that promises the benefits of improved exploration and temporal abstraction (Nachum et al., 2019). Due to this, multiple hierarchical approaches have been studied in literature (Sutton et al., 1999; Barto and Mahadevan, 2003; Parr and Russell, 1998; Dieterich, 1999). We consider a goal-conditioned setup in this work,

where a higher-level policy provides subgoals to a lower-level policy, and the lower-level policy executes primitive actions directly on the environment. In this setup, multiple prior approaches have been proposed (Dayan and Hinton, 1992; Vezhnevets et al., 2017). Although it promises these intuitive benefits, HRL has been cursed with multiple issues like non-stationarity in off-policy HRL, when multiple levels are trained simultaneously. Concretely, due to continuously changing lower-level primitive behavior, the higher-level replay buffer experience is rendered obsolete. Some prior works deal with this issue by either simulating an optimal lower-level primitive (Levy et al., 2018), or relabeling replay buffer transitions using a maximum likelihood-based approach (Nachum et al., 2018; Singh et al., 2024). In contrast, we deal with non-stationarity by using preference-based learning (Christiano et al., 2017; Lee et al., 2021). Concretely, we first derive a primitive-regularized preference-based objective, and then directly optimize the higher-level policy by employing direct preference optimization (Rafailov et al., 2024b). Some other approaches use hand-designed action or behavior priors to boost downstream learning (Nasiriany et al., 2021; Dalal et al., 2021). While such approaches effectively simplify the learning process, performance in these approaches depends on the quality of the designed priors. If such priors are sub-optimal, the learning algorithm fails to show good performance. Another line of work uses the option learning framework (Sutton et al., 1999; Klissarov et al., 2017) to learn extended macro actions. However, such approaches may lead to degenerate solutions in the absence of suitable regularization. In contrast, our approach uses primitive-enabled regularization for conditioning the higher-level policy to produce feasible subgoals, thus avoiding such degenerate solutions.

**Preference-based Learning.** In this line of work, various approaches have been proposed that perform reinforcement leaning (RL) on human preference data (Knox and Stone, 2009; Pilarski et al., 2011; Wilson et al., 2012b; Daniel et al., 2015). Prior approaches first collect preference data from human annotators, then use this data for downstream learning. An important initial work in this area is (Christiano et al., 2017), which first trains a reward model using the preference data, then uses RL to learn an optimal policy for the resulting reward model. Other recent work uses more sample-efficient off-policy policy gradient approaches (Haarnoja et al., 2018) for learning the policy. Recently, direct preference optimization approach has been proposed (Rafailov et al., 2024b;a) that circumvents the reward model learning step, by directly optimizing the policy using a KL-regularized maximum likelihood objective. In this work, we propose a novel reference policy, which directly optimizes the higher-level policy to generate feasible subgoals for lower-level policy.

## 3 PROBLEM FORMULATION

In this paper, we consider the Markov decision process (MDP) $(\mathcal{S}, \mathcal{A}, p, r, \gamma)$ framework, where $\mathcal{S}$ is the state space, $\mathcal{A}$ is the action space, $p : \mathcal{S} \times \mathcal{A} \to \Delta(\mathcal{S})$ is the transition probability function mapping state-action pairs to probability distributions over the state space, $r : \mathcal{S} \times \mathcal{A} \to \mathbb{R}$ is the reward function, and $\gamma \in (0, 1)$ is a discount factor. At timestep $t$, the agent is in state $s_t$, takes action $a_t \sim \pi(\cdot|s_t)$ according to some policy $\pi : \mathcal{S} \to \Delta(\mathcal{A})$ mapping states to probability distributions over the action space, receives reward $r_t = r(s_t, a_t)$, and the system transitions to a new state $s_{t+1} \sim p(\cdot|s_t, a_t)$. In the standard RL setting, the goal is to optimize the following objective:

$$\pi^* := \arg\max_{\pi} J(\pi) = \mathbb{E}_\pi \left[ \sum_{t=0}^{\infty} \gamma^t r_t \right]. \tag{1}$$

In what follows, we will consider the standard goal-conditioned setting (Andrychowicz et al., 2017), where the agent policy is jointly conditioned on the current state as well as a desired goal. Concretely, at timestep $t$, the policy $\pi$ predicts actions $a_t \sim \pi(\cdot|s_t, g_t)$ conditioned on both state $s_t$ and goal $g_t$. Finally, the value function for a policy $\pi$ provides the expected cumulative reward when the start state is $s_t$ and goal is $g_t$ such that $V_\pi(s_t, g_t) = \mathbb{E}_\pi[\sum_{t=0}^{T} \gamma^t r_t | s_t, g_t]$.

### 3.1 HIERARCHICAL REINFORCEMENT LEARNING

In our goal-conditioned hierarchical setup, in order to achieve the end goal, the higher-level policy provides subgoals to the lower-level policy, while the lower-level policy takes primitive actions oriented towards achieving the specified subgoals. Concretely, the higher-level policy $\pi^H : \mathcal{S} \to \Delta(\mathcal{G})$ specifies a subgoal $g_t \in \mathcal{G}$, where $\mathcal{G} \subset \mathcal{S}$ is the set of possible goals. The higher-level policy

predicts subgoal $g_t \sim \pi^H(\cdot|s_t)$ after every $k$ timesteps and $g_t = g_{k \cdot \lceil t/k \rceil}$, otherwise. Thus, the higher-level policy issues new subgoals every $k$ timesteps and keeps subgoals fixed in the interim.

Furthermore, at each $t$, the lower-level policy $\pi^L : \mathcal{S} \times \mathcal{G} \to \Delta(\mathcal{A})$ selects primitive actions $a_t \sim \pi^L(\cdot|s_t, g_t)$ according to the current state and subgoal specified by $\pi^H$, and the state transitions to $s_{t+1} \sim p(\cdot|s_t, a_t)$. Finally, the higher-level policy provides the lower level with reward $r_t^L = r^L(s_t, g_t, a_t) = -\mathbf{1}_{\{\|s_t - g_t\|_2 > \varepsilon\}}$, where $\mathbf{1}_B$ is the indicator function on a given set $B$. In the standard HRL setup where both hierarchical levels are simultaneously trained, the higher level receives reward $r_t^H = r^H(s_t, g^*, g_t)$, where $g^* \in \mathcal{G}$ is the end goal and $r^H : \mathcal{S} \times \mathcal{G} \times \mathcal{G} \to \mathbb{R}$ is the higher-level reward function. The lower level populates its replay buffer with samples of the form $(s_t, g_t, a_t, r_t^L, s_{t+1})$ after each timestep, whereas the higher level populates its buffer with samples of the form $(s_t, g^*, g_t, \sum_{i=t}^{t+k-1} r_i^H, s_{t+k})$ after $k$ timesteps. Next, we highlight key limitations of standard HRL methods.

### 3.1.1 LIMITATIONS OF STANDARD HRL APPROACHES

Although HRL promises significant advantages over non-hierarchical RL, such as improvements in sample efficiency due to temporal abstraction and improved exploration (Nachum et al., 2018; 2019), it suffers from serious limitations. In this work, we focus on two outstanding issues:

**L1:** *training instability due to lower-level non-stationarity in off-policy HRL;*
**L2:** *performance degradation due to infeasible subgoal generation by higher-level policy.*

As discussed in (Nachum et al., 2018) and (Levy et al., 2018), off-policy HRL suffers from non-stationarity due to non-stationary lower primitive behavior generated by the lower-level policy. Concretely, the higher-level replay transitions collected using previous lower-level policy become obsolete as the lower-level policy changes. Additionally, the higher level may predict infeasible subgoals to the lower-level policy (Chane-Sane et al., 2021), thus impeding learning and degrading overall performance. Hence, although standard HRL provides significant advantages, it often demonstrates poor performance in practice (Nachum et al., 2018; Levy et al., 2018; Chane-Sane et al., 2021). An important motivation of this work is to develop a novel preference-based learning approach that directly optimizes preference-based data to address the limitations mentioned above.

### 3.2 CLASSICAL RLHF METHODS

In reinforcement learning from human feedback (RLHF) (Wilson et al., 2012a; Christiano et al., 2017; Lee et al., 2021; Ibarz et al., 2018), the agent first learns a reward model using human preference feedback, then learns a policy using RL that is optimal for the resulting reward model, typically via a policy gradient method such as PPO (Schulman et al., 2017).

In this setting, the agent behavior over a $k$-length trajectory is represented as a sequence, $\tau$, of state observations and actions: $\tau = ((s_t, a_t), (s_{t+1}, a_{t+1})...(s_{t+k-1}, a_{t+k-1}))$. The reward model to be learned is represented as $\widehat{r}_\phi : \mathcal{S} \times \mathcal{A} \to \mathbb{R}$, with parameters $\phi$. Accordingly, the preferences between any two trajectories $\tau^1, \tau^2$ can be modeled using the Bradley-Terry model (Bradley and Terry, 1952):

$$P_\phi\left[\tau^1 \succ \tau^2\right] = \frac{\exp \sum_t \widehat{r}_\phi\left(s_t^1, a_t^1\right)}{\sum_{i \in \{1,2\}} \exp \sum_t \widehat{r}_\phi\left(s_t^i, a_t^i\right)}, \tag{2}$$

where $\tau^1 \succ \tau^2$ implies that $\tau^1$ is preferred over $\tau^2$. We consider the preference dataset $\mathcal{D}$ with entries of the form $(\tau^1, \tau^2, y)$, where $y = (1, 0)$ when $\tau^1$ is preferred over $\tau^2$, $y = (0, 1)$ when $\tau^2$ is preferred over $\tau^1$, and $y = (0.5, 0.5)$ when there is no preference. The standard approach in the preference-based literature (see (Christiano et al., 2017; Lee et al., 2021)) is to learn the reward function $\widehat{r}_\phi$ using the following cross-entropy loss:

$$\mathrm{L}(\phi) = -\sum_{\mathcal{D}} \left(y_1 \log P_\phi\left[\tau^1 \succ \tau^2\right] + y_2 \log P_\phi\left[\tau^2 \succ \tau^1\right]\right), \tag{3}$$

where $(\tau^1, \tau^2, y) \in \mathcal{D}$ and $y = [y_1, y_2]$.

## 3.3 DIRECT PREFERENCE OPTIMIZATION

Unlike classical RLHF, direct preference optimization (DPO) circumvents the need for an RL algorithm by using a closed-form solution for the optimal policy of the KL-regularized RL problem (Levine, 2018; Ziebart et al., 2008), which takes the form $\pi^*(a|s) = \frac{1}{Z(s)}\pi_{ref}(a|s)e^{r(s,a)}$, where $\pi_{ref}$ is the reference policy, $\pi^*$ is the optimal policy, and $Z(s)$ is a normalizing partition function ensuring that $\pi^*$ provides a valid probability distribution over $\mathcal{A}$ for each $s \in \mathcal{S}$. This formulation is rearranged to yield an alternative expression $r(s,a) = \alpha \log \pi^*(a|s) - \alpha \log \pi_{ref}(a|s) - Z(s)$ for the reward function. This equation is then substituted in the standard cross-entropy loss equation 3, which yields the following objective (Rafailov et al., 2024b):

$$\mathcal{L}_{DPO} = -\mathbb{E}_{(s,y_1,y_2)\sim\mathbb{D}} \left[ \log \sigma \left( \alpha \log \frac{\pi_\theta(y_1|s)}{\pi_{ref}(y_1|s)} - \alpha \log \frac{\pi_\theta(y_2|s)}{\pi_{ref}(y_2|s)} \right) \right] \tag{4}$$

where $\theta$ are the policy parameters and $\sigma(x) = (1 + e^{-x})^{-1}$ denotes the sigmoid function.

## 4 PROPOSED APPROACH

In this section, we introduce DIPPER: **DI**rect **P**reference Optimization to Accelerate **P**rimitive-**E**nabled Hierarchical **R**einforcement Learning. To address the problem of learning control problems for complex robotics tasks from human preference data, a natural approach is to apply a combination of RLHF and HRL: on the outer highest tier, a reward model is learned from the human preference data, on the middle tier RL is used to learn a corresponding higher-level policy for subgoal generation, and on the third, lowest tier RL is used to learn lower-level policies for achieving subgoals specified by the higher-level policy. Together, the lower and middle tiers in this approach naturally correspond to performing RLHF, while the middle and higher tiers correspond to performing HRL. Though intuitively reasonable, the need to carry out three distinct learning procedures simultaneously in this approach is computationally burdensome and a more efficient method is required.

**Our key idea.** The key idea underlying DIPPER is twofold: we introduce a DPO-based approach to directly learn higher-level policies from preferences, replacing the two-tier RLHF component in the scheme described above with a simpler, more efficient single-tier approach; we replace the reference policy inherent in DPO-based approaches, which is typically unavailable in complex robotics tasks, with a primitive-enabled reference policy derived from a novel bi-level optimization formulation of the HRL problem. The result is an efficient hierarchical approach that directly optimizes the higher-level policy using preference data while simultaneously mitigating non-stationarity and infeasible subgoal prediction issues of HRL (see Section 3.1.1) through primitive-enabled regularization.

### 4.1 DIPPER

We now introduce our hierarchical approach DIPPER, which uses a primitive-enabled direct preference optimization formulation to optimize the higher-level policy and RL to optimize the lower-level policy. Recalling the HRL and RLHF settings of Sections 3.1 and 3.2, let $V_{\pi_L}(s_t, g_t)$ denote the lower-level value function and $r_\phi$ denote a parameterized reward model corresponding to the preference data. In addition, let $\alpha \geq 0$ be a scalar hyperparameter controlling the magnitude of the KL-regularization term between higher level policy $\pi_U$ and reference policy $\pi_{ref}$. For a trajectory $\tau$ of length $T$, we consider the following KL-regularized optimization problem:

$$\max_{\pi_U} \mathbb{E}_{\pi_U} \left[ \sum_{t=0}^{T} \left( r_\phi(s_t, g_t) - \alpha \mathbb{D}_{\text{KL}}[\pi_U(\cdot|s_t) \| \pi_{ref}(\cdot|s_t)] \right) \right], \tag{5}$$

In the standard DPO setting considered in (Rafailov et al., 2024b;a), the reference policy $\pi_{ref}$ is assumed to be given. In challenging problems such as the robotics tasks motivating this work, however, such a reference policy is often unavailable. We must therefore seek an alternative reference policy corresponding to the HRL problem at hand. In order to achieve this, we next provide a novel bi-level formulation of the HRL problem that we subsequently leverage to propose a suitable $\pi_{ref}$.

### 4.1.1 BI-LEVEL OPTIMIZATION FORMULATION OF HRL

We now present our bi-level optimization formulation of the HRL problem. For a given higher-level policy $\pi_U$, let $\pi_L^*$ denote the corresponding optimal lower-level policy. Let $\tau =$

$((s_t, g_t), (s_{t+1}, g_{t+1})...(s_{t+k-1}, g_{t+k-1}))$ represent the higher-level trajectories, where $s_t$ is the State at time $t$, and $g_t \sim \pi_U(.|s_t, g^*)$ is the subgoal predicted by the higher-level policy at time $t$. Notably, the higher-level policy $\pi_U$ predicts the subgoal $g_t$ for the lower-level policy, which is kept fixed for $k$ timesteps while the lower level policy $\pi_L^*$ executes. Hence, the next state $s_{t+1}$ depends on the optimal lower-level policy $\pi_L^*$. We represent our hierarchical learning problem as the following bi-level optimization problem:

$$\max_{\pi_U} \mathcal{J}(\pi_U, \pi_L^*(\pi_U)) \quad s.t. \quad \pi_L^*(\pi_U) = \arg\max_{\pi_L} V_{\pi_L}(\pi_U), \tag{6}$$

where $\mathcal{J}(\pi_U, \pi_L^*(\pi_U))$ represents the higher level maximization objective, and $V_{\pi^L}(\pi^H)$ is the lower level value function, conditioned on the higher level policy subgoals. Note that, in the given constraint, the optimal lower-level policy $\pi_L^*$ is defined as the policy which maximizes the lower-level value function $V_{\pi_L}$. We can solve this bi-level joint optimization for the higher-level policy. In order to optimize for both $\pi_U$ and $\pi_L$, we can reformulate equation 6 as follows (Liu et al., 2022):

$$\max_{\pi_U, \pi_L} \mathcal{J}(\pi_U, \pi_L) \quad s.t. \quad V_{\pi_L}(\pi_U) - V_{\pi_L}^*(\pi_U) \geq 0, \tag{7}$$

where, $V_{\pi_L}^*(\pi_U) = \max_{\pi_L} V_{\pi_L}(\pi_U)$. Notably, since the left-hand side of the inequality constraint is always non-positive due to the fact that $V_{\pi_L}(\pi_U) - V_{\pi_L}^*(\pi_U) \leq 0$, the constraint is satisfied only when $V_{\pi_L}(\pi_U) = V_{\pi_L}^*(\pi_U)$. Finally, equation 7 can be formulated as the following Lagrangian:

$$\max_{\pi_U, \pi_L} \mathcal{J}(\pi_U, \pi_L) + \lambda(V_{\pi_L}(\pi_U) - V_{\pi_L}^*(\pi_U)). \tag{8}$$

We now use the formulation of `HRL` in equation 8 to propose a novel reference policy for our `DPO`-based objective. This yields an efficient `HRL` algorithm dealing with non-stationarity and infeasible subgoal generation that is able to solve complex robotics tasks (cf. Section 5).

### 4.1.2 `DIPPER` Reference Policies

A key component of the `DPO`-based approach is to provide a suitable reference policy (cf. equation 4), which is difficult to obtain in the robtics tasks. In light of the regularized objective equation 8 derived in Section 4.1.1, we propose the following formulation of the reference policy:

$$\pi_{ref}(g_t|s_t) = \frac{\exp(m(V_{\pi_L}(s_t, g_t) - V_{\pi_L}^*(s_t, g_t)))}{Z(s_t)}, \tag{9}$$

where $Z(s_t) = \sum_{g_t} \exp\left(m(V_{\pi_L}(s_t, g_t) - V_{\pi_L}^*(s_t, g_t))\right)$, $V_{\pi_L}^*(s_t, g_t) = \max_{\pi_L} V_{\pi_L}(s_t, g_t)$, and $m = \frac{\lambda}{\alpha}$. Note that, since the term $V_{\pi_L}(s_t, g_t) - V_{\pi_L}^*(s_t, g_t)$ in the numerator is always non-positive, for a given $g_t$, the term is maximized when $V_{\pi_L}(s_t, g_t) = V_{\pi_L}^*(s_t, g_t)$. Equivalently, the term is maximized when, for a particular $g_t$, the lower-level value function is optimal. We show later on in Section 4.1.3 that, when this particular choice of reference policy is substituted in the `DPO` objective, we get exactly the formulation in equation 8.

In addition to its connections to the bi-level formulation, the specific form of the reference policy leads to significant advantages with respect to the hierarchical component of our approach. To see this, notice that the reference policy $\pi_{ref}(g_t|s_t)$ assigns high probability to the subgoal $g_t$, where the corresponding lower-level value function $V_{\pi_L}(s_t, g_t)$ is close to optimal, or alternatively, where the corresponding lower-level policy $\pi_L(s_t, g_t)$ is close to optimal. This formulation effectively handles the non-stationarity issue (**L1**) and infeasible subgoal generation issue (**L2**) in `HRL` as follows:

**Dealing with L1**: For a particular subgoal $g_t$, if the lower-level policy is close to optimal, it predicts actions similar to the optimal lower-level policy. This reduces the non-stationary behavior of the lower-level policy, which ameliorates the non-stationarity issue in `HRL`.

**Dealing with L2**: For a state $s_t$ and subgoal $g_t$, $V_{\pi_L}(s_t, g_t)$ provides an estimate of the feasibility of subgoal $g_t$, since a high value of $V_{\pi_L}(s_t, g_t)$ implies that the lower level expects to achieve high reward for subgoal $g_t$. Since $\pi_{ref}$ assigns high probability to subgoals with large $V_{\pi_L}(s_t, g_t)$, $\pi_{ref}$ produces achievable subgoals, thus mitigating infeasible subgoal generation issue in `HRL`.

### 4.1.3 DIPPER OBJECTIVE

Here, we derive our DIPPER objective. We first substitute the proposed reference policy of equation 9 into the DPO objective equation 5 to get the following formulation:

$$\max_{\pi_U} \mathbb{E}_{\pi_U} \left[ \sum_{t=0}^{T} (r_\phi(s_t, g_t) + \lambda(V_{\pi_L}(s_t, g_t) - V_{\pi_L}^*(s_t, g_t)) + \hat{m}(s_t)) \right], \tag{10}$$

where $\hat{m}(s_t) = (\alpha \mathcal{H}(s_t) - \alpha \log Z(s_t))$, and $\mathcal{H}(s_t) = -\log \pi_U(g_t|s_t)$ is the entropy term for higher-level policy. When optimizing for the higher-level policy, we can choose to ignore the term $\hat{m}(s_t)$, since it does not depend on the policy $\pi_U(g_t|s_t)$. Note that the formulation in equation 10 is exactly equal to the bi-level formulation of equation 8. Hence, when we plug in the proposed form of reference policyequation 4.1.2 in KL-regularized DPO objective, this yields the formulation in equation 8. Following prior works (Levine, 2018; Ziebart et al., 2008) and substituting the reference policy in equation 5, we get the following optimal solution for the higher-level policy:

$$\pi_U(g_t|s_t) = \frac{1}{Z(s_t)} \exp(\frac{1}{\alpha}(r_\phi(s_t, g_t) + \lambda(V_{\pi_L}(s_t, g_t) - V_{\pi_L}^*(s_t, g_t)))), \tag{11}$$

where $Z(s_t) = \sum_{g_t} \exp(\frac{1}{\alpha}(r_\phi(s_t, g_t) + \lambda(V_{\pi_L}(s_t, g_t) - V_{\pi_L}^*(s_t, g_t))))$ is the partition function and $\lambda$ is the primitive regularization weight hyper-parameter. Appendix A.1 contains a complete derivation. Taking logarithm on both sides of equation 11 and using some basic algebra yields:

$$r_\phi(s_t, g_t) = \alpha \log Z(s_t) + \alpha \log \pi_U(g_t|s_t) - \lambda(V_{\pi_L}(s_t, g_t) - V_{\pi_L}^*(s_t, g_t)). \tag{12}$$

We can reformulate the Bradley-Terry model (Bradley and Terry, 1952) to derive the following:

$$\mathcal{L}^d = -\mathbb{E}_{(\tau_1, \tau_2) \sim \mathbb{D}} \left[ \log \sigma \left( \sum_{t=0}^{T} r_\phi(s_t^1, g_t^1) - \sum_{t=0}^{T} r_\phi(s_t^2, g_t^2) \right) \right]. \tag{13}$$

We now substitute the preference reward formulation equation 12 into equation 13 to derive our final maximum likelihood objective:

$$\mathcal{L}^d = -\mathbb{E}_{(\tau_1, \tau_2) \sim \mathbb{D}} \left[ \log \sigma (\sum_{t=0}^{T} (\alpha \log \pi_U(g_t^1|s_t^1) - \alpha \log \pi_U(g_t^2|s_t^2)) \right. \tag{14}$$
$$\left. - \lambda((V_{\pi_L}(s_t^1, g_t^1) - V_{\pi_L}^*(s_t^1, g_t^1)) - (V_{\pi_L}(s_t^2, g_t^2) - V_{\pi_L}^*(s_t^2, g_t^2))) \right].$$

This objective provides the maximum likelihood DIPPER objective for optimizing the higher-level policy $\pi_U$, while also using uses primitive-enabled regularization that regularizes the higher level policy to predict feasible subgoals for the lower-level policy.

### 4.1.4 DIPPER: A PRACTICAL ALGORITHM

We now employ the derived DIPPER formulation to propose an efficient and practically applicable DPO-based algorithm. Notably, equation 14 requires calculation of optimal value function $V_{\pi_L}^*(s_t, g_t)$ for a subgoal $g_t$. Unfortunately, computing optimal value functions is computationally expensive and is typically not practically feasible. We accordingly consider an approximation $V_{\pi_L}^k(s_t, g_t)$ to replace $V_{\pi_L}^*(s_t, g_t)$, where $k$ represents the number of training iterations for updating $V_L^k(s_t, g_t)$. Further, we make an approximation and ignore the term $V_{\pi_L}$ in equation 14. We explain our rationale to ignore $V_{\pi_L}$ as follows: without loss of generality, let us assume that the environment rewards are greater than and equal to zero. This directly implies that $V_{\pi_L} \geq 0$. We utilize this to maximize the lower bound of objective in equation 10, and follow similar steps between equation 10 to equation 14, to present the final practically applicable maximum likelihood DIPPER objective:

$$\mathcal{L}^d = -\mathbb{E}_{(\tau_1, \tau_2) \sim \mathbb{D}}[\log \sigma(\sum_{t=0}^{T} (\alpha \log \pi_U(g_t^1|s_t^1) - \alpha \log \pi_U(g_t^2|s_t^2)) + \lambda(V_L^k(s_t^1, g_t^1) - V_L^k(s_t^2, g_t^2))].$$
$$\tag{15}$$

We note that the objective in equation 15 still captures the core essense of the proposed approach and tries to deal with the non-stationarity issue in HRL and also learn a lower level regularized

upper level policy to deal with infeasible subgoal geneation. Despite these approximations, in our experiments we empirically find that DIPPER is able to efficiently mitigate the recurring issue of non-stationarity in HRL and generate feasible subgoals for the lower-level policy.

**Analyzing DIPPER gradient:** We further analyze the rationale behind the DIPPER objective by computing and interpreting the gradients of $\mathcal{L}^d$ with respect to higher level policy $\pi_U$. The gradient can be written as:

$$\nabla\mathcal{L}^d = -\alpha\mathbb{E}_{(\tau_1,\tau_2)\sim\mathbb{D}}\left[\sum_{t=0}^{T}\underbrace{(\sigma(\hat{r}(s_t^2,g_t^2)-\hat{r}(s_t^1,g_t^1))}_{\text{higher weight for wrong preference}}*[\underbrace{\nabla\log\pi_U(g_t^1|s_t^1)}_{\text{increase likelihood of }\tau_1}-\underbrace{\nabla\log\pi_U(g_t^2|s_t^2)]}_{\text{decrease likelihood of }\tau_2})\right]$$
(16)

where $\hat{r}(s_t,g_t)=\alpha\log\pi_U(g_t|s_t)-\lambda(V_{\pi_L}(s_t,g_t)-V_{\pi_L}^*(s_t,g_t))$ is the implicit reward defined by the higher-level policy and lower-level value function. Intuitively, the gradient increases the likelihood of preferred trajectories and decreases the likelihood of dispreferred ones. The gradient difference is weighted by how incorrectly the implicit reward model $\hat{r}(s_t,g_t)$ orders the trajectories, according to the strength of the KL constraint. We provide DIPPER algorithm in Appendix A.3 Algorithm 1.

## 5 EXPERIMENTS

In this section, we perform extensive empirical analysis, and ask the following questions: **(1)** Does DIPPER enhance sample efficiency and training stability in complex robotic manipulation and navigation tasks, compared to the baselines? **(2)** Is DIPPER able to mitigate the recurring issue of non-stationarity in HRL? **(3)** Is DIPPER able to generate feasible subgoals for the lower primitive? **(4)** What is the contribution of each of our design choices?

**Setup.** We evaluate DIPPER on four robotic navigation and manipulation tasks: $(i)$ maze navigation, $(ii)$ pick and place (Andrychowicz et al., 2017), $(iii)$ push, and $(iv)$ franka kitchen (Gupta et al., 2019). These are sparse reward enviroments, where the lower primitive is sparsely rewarded when it comes within $\delta$ distance of the subgoal. Unless explicitly stated, we ensure fair comparisons across all the baselines. Notably, since the pick and place, push and kitchen task environments are complex sparse reward environments, we assume access to a single human demonstration, and use an additional imitation learning objective at the lower level. We do not assume access to any demonstration in the maze navigation task. This is done to speedup training, however, we keep this assumption consistent among all baselines to ascertain fair comparisons. We provide additional implementation details in Appendix A.5, and the implementation code in the supplementary.

### 5.1 EVALUATION AND RESULTS.

Here, we compare the success rate performances on four sparse maze navigation and robotic manipulation tasks in Figure 2. The solid line and shaded regions represent the mean and standard deviation, averaged over 5 seeds.

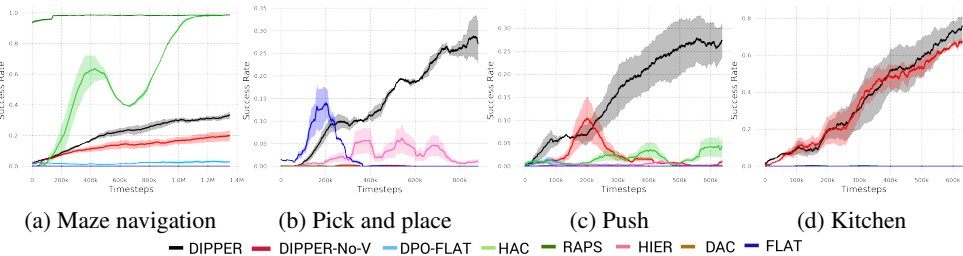

(a) Maze navigation  (b) Pick and place  (c) Push  (d) Kitchen

— DIPPER — DIPPER-No-V — DPO-FLAT — HAC — RAPS — HIER — DAC — FLAT

Figure 2: **Performance comparison** This figure compares the success rate performances on four sparse maze navigation and robotic manipulation tasks. The solid line and shaded regions represent the mean and standard deviation, across 5 seeds. We compare DIPPER against multiple baselines. Although HAC and RAPS outperform DIPPER in simpler maze task, DIPPER significantly outperforms the baselines in harder tasks.

### 5.1.1 COMPARING WITH DPO BASELINES

Here we compare DIPPER against DPO based baselines, specifically: $(i)$ DIPPER-No-V (DIPPER without primitive-enabled regularization), and $(ii)$ DPO-FLAT (Single-level DPO implementation).

**DIPPER-No-V:** In order to illustrate the importance of primitive regularization employing lower primitive value function, we implement DIPPER-No-V baseline by removing primitive regularization from DIPPER. As seen from Figure 2, DIPPER performs slightly better than DIPPER-No-V in simpler maze navigation task and in kitchen task. However, DIPPER significantly outperforms DIPPER-No-V baseline in pick and place and push tasks. This clearly demonstrates the advantage of primitive regularization, which conditions the higher level policy to predict feasible subgoals.

**DPO-FLAT:** DPO-Flat is a single-level implementation of DPO (Rafailov et al., 2024b). We implemented this baseline to illustrate that our hierarchical DPO based approach (where the higher policy is trained using DPO based maximum likelihood objective, and the lower policy is trained using RL) outperforms single-level DPO based policy. Since DIPPER is hierarchical, it benefits from factors like temporal abstraction and improved exploration, which are missing from single-level DPO implementation. However, since we do not have access to a reference policy in robotics, we replace the reference policy with a uniform policy. Notably, this particular choice of reference policy effectively reformulates the KL-objective into an entropy maximization objective in DPO-Flat, which facilitates better exploration. DIPPER clearly outperforms this baseline in all the tasks, showing that our hierarchical structure is crucial for improved performance.

### 5.1.2 COMPARING WITH HIERARCHICAL BASELINES

Here we compare DIPPER against hierarchical baselines, specifically: $(i)$ RAPS (Dalal et al., 2021), $(ii)$ HAC (Levy et al., 2018), and $(iii)$ HIER (vanilla hierarchical SAC implementation).

**RAPS:** We compare DIPPER with RAPS baseline to analyze how DIPPER performs against approaches that use behavior priors or action primitives. Notably, the performance of RAPS depends on the quality of such priors, and require considerable effort to hand-design, especially in hard environments like franka kitchen. We find that RAPS is able to significantly outperform DIPPER in maze task, which we believe is because the designed action primitive in maze task is near perfect. However, as the complexity of environments increase, RAPS is unable to show any progress.

**HAC:** We also implemented Hierarchical Actor Critic (Levy et al., 2018), which deals with non-stationarity in HRL by simulating optimal lower primitive behavior. As seen in Figure 2, HAC is able to outperform DIPPER in simpler maze navigation task. However, in harder pick and place, push and kitchen tasks, DIPPER significantly outperforms this baseline.

**HIER:** HIER is a vanilla HRL baseline implemented using SAC (Haarnoja et al., 2018). However, this baseline failed to perform well, especially in complex tasks.

### 5.1.3 COMPARING WITH NON-HIERARCHICAL BASELINES

Here we compare DIPPER against non-hierarchical baselines, specifically $(i)$ DAC (Discriminator Actor Critic (Kostrikov et al., 2018)), and $(ii)$ FLAT (Single-level SAC (Haarnoja et al., 2018)).

**DAC:** We provide one demonstration to DAC baseline in each environment. However, as seen in Figure 2, even with privileged information, DAC is unable to perform well.

**FLAT:** As seen in Figure 2, FLAT baseline is unable to perform well in any of the tasks, highlighting the importance of our hierarchical structure for success in complex robotic tasks.

### 5.2 ABLATION ANALYSIS

Here, we perform various ablations to analyze the contribution of each of our design choices.

**Dealing with non-stationarity in HRL:** We evaluate whether DIPPER reduces non-stationarity in HRL by comparing it to the vanilla HIER baseline, as shown in Figure 3. We measure the average distance between subgoals predicted by the higher-level policy and those achieved by the lower-level primitive at different stages of training. A low average distance indicates that DIPPER effectively predicts subgoals achievable by the lower primitive, thus promoting optimal lower primitive be-

havior. Our results show that DIPPER consistently produces low average distances, confirming its ability to mitigate non-stationarity.

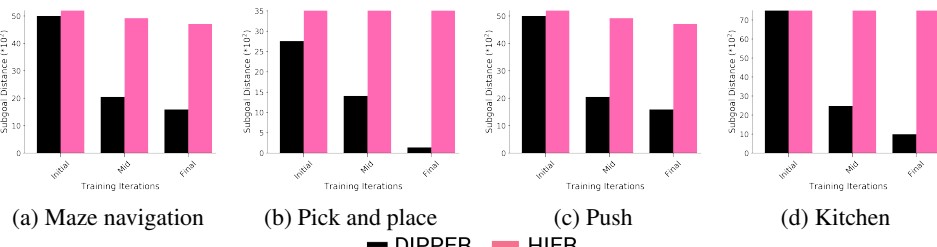

(a) Maze navigation     (b) Pick and place     (c) Push     (d) Kitchen

Figure 3: **Non-stationarity metric comparison** This figure compares DIPPER with the HIER baseline according to the average distance between subgoals proposed by the higher-level policy and the subgoals achieved by the lower-level primitive *throughout the training process*. DIPPER consistently demonstrates lower average distance values, which implies that DIPPER higher-level policy predicts feasible subgoals, inducing optimal lower level primitive behavior, thereby leading to non-stationarity mitigation and enhanced task performance.

**Dealing with infeasible subgoal generation in HRL:** In Figure 4, we compare DIPPER with the HIER baseline by evaluating the average distance between subgoals predicted by the higher-level policy and those achieved by the lower-level policy *after training is completed*. As seen in Figure 4, the distance values for DIPPER are significantly lower than those of the HIER baseline, indicating that DIPPER generates feasible subgoals by exploiting primitive regularization.

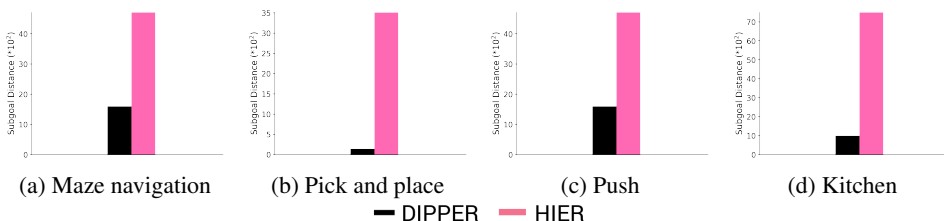

(a) Maze navigation     (b) Pick and place     (c) Push     (d) Kitchen

Figure 4: **Feasible subgoal generation metric comparison** This figure compares DIPPER with the HIER baseline using the average distance between subgoals predicted by the higher-level policy and those achieved by the lower-level policy *after training is completed*. DIPPER exhibits significantly lower average distance values compared to HIER baseline, showing that DIPPER generates feasible subgoals for the lower-level primitive.

**Additional Ablations:** We compare the success rate performance of DIPPER against DIPPER-Random, which is DIPPER implemented with a random reference policy. This baseline is used to demonstrate the significance of primitive regularization induced by our novel formulation of reference policy. As can be seen in Appendix A.4 Figure 5, DIPPER significantly outperforms this baseline on all tasks, which shows that primitive regularization is crucial for enhanced performance. We also perform ablation studies and intuitions for selecting the primitive regularization weight hyper-parameter $\lambda$ and the KL regularization weight $\alpha$ in Appendix A.4 Figures 6 and 7.

# 6 CONCLUSION

In this work, we propose DIPPER, a preference learning based HRL algorithm that employs direct policy optimization and primitive enabled regularization to mitigate the issues of non-stationarity and infeasible subgoal generation in HRL. We employ a bi-level optimization formulation for HRL and use it to propose a novel reference policy formulation which results in our primitive regularized maximum likelihood objective. We empirically show that DIPPER demonstrates impressive performance on complex robotic control tasks, and is able to significantly outperform the baselines. Additionally, our hierarchical formulation outperforms single level DPO formulation. Based on our strong empirical findings, we believe that DIPPER represents a significant advancement in developing effective control policies for addressing complex, sparse-reward robotic tasks. Due to space limit, we discuss the limitations and future work in Appendix A.6.

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

CONTENTS

# A APPENDIX

## A.1 DERIVING THE FINAL OPTIMUM OF KL-CONSTRAINED REWARD MAXIMIZATION OBJECTIVE

In this appendix, we will derive Eqn 11 from Eqn 5. Thus, we optimize the following objective:

$$\mathsf{P} := \max_{\pi_U} \mathbb{E}_{\pi_U}[\sum_{t=0}^{T}(r_\phi(s_t, g_t) - \alpha \mathbb{D}_{\mathrm{KL}}[\pi_U(\cdot|s_t)\|\pi_{ref}(\cdot|s_t)])], \tag{17}$$

Re-writing the above equation after expanding KL divergence formula:

$$\mathsf{P} = \max_{\pi_U} \mathbb{E}_{\pi_U}[\sum_{t=0}^{T}(r_\phi(s_t, g_t) - \alpha \log \frac{\pi_U(g_t|s_t)}{\pi_{ref}(g_t|s_t)})] \tag{18}$$

$$= \max_{\pi_U} \mathbb{E}_{\pi_U}[\sum_{t=0}^{T}(r_\phi(s_t, g_t) - \alpha \log \pi_U(g_t|s_t) + \alpha \log \pi_{ref}(g_t|s_t))]. \tag{19}$$

Substituting $\pi_{ref}$ from Eqn 9, and $m = \frac{\lambda}{\alpha}$ in Equation 19,

$$\mathsf{P} = \max_{\pi_U} \mathbb{E}_{\pi_U}[\sum_{t=0}^{T}(r_\phi(s_t, g_t) - \alpha \log \pi_U(g_t|s_t) + \alpha \log \exp(k(V_L(s_t, g_t) - V_L^*(s_t, g_t)))$$
$$- \alpha \log \sum_{g_t} \exp(k(V_L(s_t, g_t) - V_L^*(s_t, g_t))))] \tag{20}$$

$$= \max_{\pi_U} \mathbb{E}_{\pi_U}[\sum_{t=0}^{T}(r_\phi(s_t, g_t) - \alpha \log \pi_U(g_t|s_t) + \lambda(V_L(s_t, g_t) - V_L^*(s_t, g_t))$$
$$- \alpha \log \sum_{g_t} \exp(k(V_L(s_t, g_t) - V_L^*(s_t, g_t))))] \tag{21}$$

$$= \min_{\pi_U} \mathbb{E}_{\pi_U}[\sum_{t=0}^{T}(\log \pi_U(g_t|s_t) - \frac{1}{\alpha}(r_\phi(s_t, g_t) + \lambda(V_L(s_t, g_t) - V_L^*(s_t, g_t)))$$
$$+ \log \sum_{g_t} \exp(k(V_L(s_t, g_t) - V_L^*(s_t, g_t))))] \tag{22}$$

$$= \min_{\pi_U} \mathbb{E}_{\pi_U}[\sum_{t=0}^{T}(\log(\frac{\pi_U(g_t|s_t)}{\exp(\frac{1}{\alpha}(r_\phi(s_t, g_t) + \lambda(V_L(s_t, g_t) - V_L^*(s_t, g_t))))})$$
$$+ \log \sum_{g_t} \exp(k(V_L(s_t, g_t) - V_L^*(s_t, g_t))))]. \tag{23}$$

After rearranging the terms, we get

$$\mathsf{P} = \min_{\pi_U} \mathbb{E}_{\pi_U}[\sum_{t=0}^{T}(\log(\frac{\pi_U(g_t|s_t)}{\frac{1}{Z(s)}\exp(\frac{1}{\alpha}(r_\phi(s_t, g_t) + \lambda(V_L(s_t, g_t) - V_L^*(s_t, g_t))))})$$
$$+ \log \sum_{g_t} \exp(k(V_L(s_t, g_t) - V_L^*(s_t, g_t))) - \log Z(s))] \tag{24}$$

where, $Z(s) = \sum_{g_t} \exp(\frac{1}{\alpha}(r_\phi(s_t, g_t) + \lambda(V_L(s_t, g_t) - V_L^*(s_t, g_t))))$.

Note that the partition function Z(s) and the term $\log \sum_{g_t} \exp(k(V_L(s_t, g_t) - V_L^*(s_t, g_t)))$, do not depend on the policy $\pi_U$

$$= \min_{\pi_U} \mathbb{E}_{\pi_U} \left[ \sum_{t=0}^{T} (\mathbb{D}_{\mathrm{KL}}[\pi_U(g_t|s_t) \| \pi_U^*(g_t|s_t)] - \log \sum_{g_t} \exp(k(V_L(s_t, g_t) - V_L^*(s_t, g_t))) - \log Z(s)) \right]$$

(25)

where, $\pi_U^*(g_t|s_t) = \frac{1}{Z(s)} \exp(\frac{1}{\alpha}(r_\phi(s_t, g_t) + \lambda(V_L(s_t, g_t) - V_L^*(s_t, g_t))))$ which is a valid probability distribution. $\pi_U^*(g_t|s_t)$ is minimized when, $D_{\mathrm{KL}} = 0$. Hence,

$$\pi_U(g_t|s_t) = \pi_U^*(g_t|s_t) = \frac{1}{Z(s)} \exp(\frac{1}{\alpha}(r_\phi(s_t, g_t) + \lambda(V_L(s_t, g_t) - V_L^*(s_t, g_t)))) \quad (26)$$

## A.2 IMPLEMENTATION DETAILS

We perform the experiments on two system each with Intel Core i7 processors, equipped with 48GB RAM and Nvidia GeForce GTX 1080 GPUs. We also provide the timesteps taken for running the experiments. For environments $(i) - (iv)$, the maximum task horizon $\mathcal{T}$ is set to 225, 50, 50, 225 timesteps, respectively, and the lower primitive is allowed to execute for $15, 7, 7$ and $15$ timesteps, respectively. In our experiments, we use off-policy Soft Actor Critic (SAC) (Haarnoja et al., 2018) for optimizing RL objective, using the Adam (Kingma and Ba, 2014) optimizer. The actor and critic networks are formulated as three-layer, fully connected neural networks with $512$ neurons in each layer. The experiments are run for 1.35e6, 9e5, 1.3E6, and 6.3e5 timesteps in environments $(i) - (iv)$, respectively. In the maze navigation task, a 7-degree-of-freedom (7-DoF) robotic arm traverses a four-room maze, with its closed gripper (fixed at table height) maneuvering through the maze to reach the goal position.

For the pick and place task, the 7-DoF robotic arm gripper must locate a square block, pick it up, and deliver it to the goal position. In the push task, the 7-DoF robotic arm gripper is required to push the square block toward the goal position. In the kitchen task, a 9-DoF Franka robot must execute a pre-defined complex task to achieve the final goal, specifically, opening the microwave door. We compare our approach to the Discriminator Actor-Critic (Kostrikov et al., 2018), which is provided with a single expert demonstration. Although not explored here, combining preference-based learning and learning from demonstrations presents an intriguing research direction (Cao et al., 2020).

To ensure fair comparisons, we maintain consistency across all baselines by keeping parameters such as neural network layer width, number of layers, choice of optimizer, SAC implementation parameters, etc., the same wherever possible. In RAPS, the lower-level behaviors are as follows: for maze navigation, we design a single primitive, *reach*, where the lower-level primitive moves in a straight line towards the subgoal predicted by the higher level. For the pick and place and push tasks, we design three primitives: *gripper-reach*, where the gripper moves to a specified position $(x_i, y_i, z_i)$; *gripper-open*, which opens the gripper; and *gripper-close*, which closes the gripper. In the kitchen task, we use the action primitives implemented in RAPS (Dalal et al., 2021).

### A.2.1 ADDITIONAL HYPER-PARAMETERS

Here, we enlist the additional hyper-parameters used in DIPPER:
**activation:** tanh [activation for reward model]
**layers:** 3 [number of layers in the critic/actor networks]
**hidden:** 512 [number of neurons in each hidden layers]
**Q_lr:** 0.001 [critic learning rate]
**pi_lr:** 0.001 [actor learning rate]
**buffer_size:** int(1E7) [for experience replay]
**clip_obs:** 200 [clip observation]
**n_cycles:** 1 [per epoch]
**n_batches:** 10 [training batches per cycle]
**batch_size:** 1024 [batch size hyper-parameter]
**reward_batch_size:** 50 [reward batch size for DPO-FLAT]

```
random_eps:  0.2 [percentage of time a random action is taken]
alpha:  0.05 [weightage parameter for SAC]
noise_eps:  0.05 [std of gaussian noise added to
not-completely-random actions]
norm_eps:  0.01 [epsilon used for observation normalization]
norm_clip:  5 [normalized observations are cropped to this values]
adam_beta1:  0.9 [beta 1 for Adam optimizer]
adam_beta2:  0.999 [beta 2 for Adam optimizer]
```

### A.3 DIPPER ALGORITHM

Here, we provide the pseudo-code for DIPPER algorithm

---

**Algorithm 1** DIPPER

---

1: Initialize preference dataset $\mathcal{D} = \{\}$
2: Initialize lower level replay buffer $\mathcal{R}^L = \{\}$
3: **for** $i = 1 \dots N$ **do**
4:     // Collect higher level trajectories $\tau$ using $\pi^H$ and lower level trajectories $\rho$ using $\pi^L$,
5:     // and store the trajectories in $\mathcal{D}$ and $\mathcal{R}^L$ respectively
6:     // After every g timesteps, relabel $\mathcal{D}$ using human preference feedback $y$
7:     // Lower level value function update
8:     **for** each gradient step in t=0 to k **do**
9:         Optimize lower level value function $V_{\pi_L}$ to get $V_{\pi_L}^k$
10:    // Higher level policy update using DIPPER
11:    **for** each gradient step **do**
12:        // Sample higher level behavior trajectories
13:        $(\tau^1, \tau^2, y) \sim \mathcal{D}$
14:        Optimize higher level policy $\pi^U$ using equation 15
15:    // Lower primitive policy update using `RL`
16:    **for** each gradient step **do**
17:        Sample $\rho$ from $\mathcal{R}^L$
18:        Optimize lower policy $\pi^L$ using SAC

---

### A.4 ADDITIONAL ABLATION EXPERIMENTS

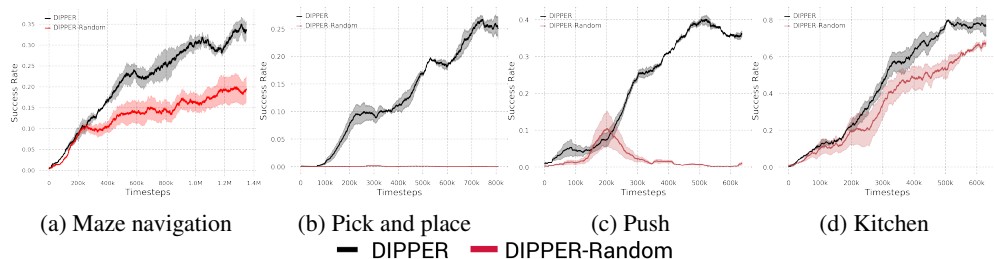

(a) Maze navigation      (b) Pick and place      (c) Push      (d) Kitchen

Figure 5: **Comparison with random reference policy** This figure compares the success rate performances of DIPPER against DIPPER-Random, which is DIPPER implemented with a random reference policy. As can be seen, DIPPER significantly outperforms DIPPER-Random, which shows that our proposed reference policy formulation demonstrates impressive performance on all tasks.

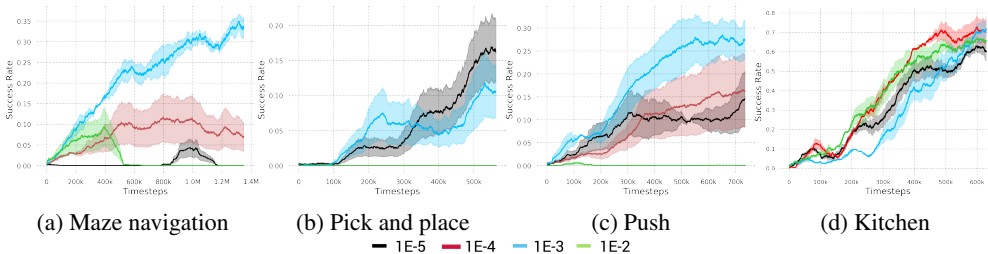

(a) Maze navigation     (b) Pick and place     (c) Push     (d) Kitchen

Figure 6: **Regularization hyper-parameter ablation** This figure compares the success rate performances for various values of primitive regularization weight $\lambda$ hyper-parameter. If $\alpha$ is too small, we loose the advantages of primitive informed regularization, leading to poor performance. In contrast, if $\alpha$ is too large, it may lead to degenerate solutions. Thus, picking proper $\lambda$ value is crucial for appropriate subgoal prediction, and improving overall performance.

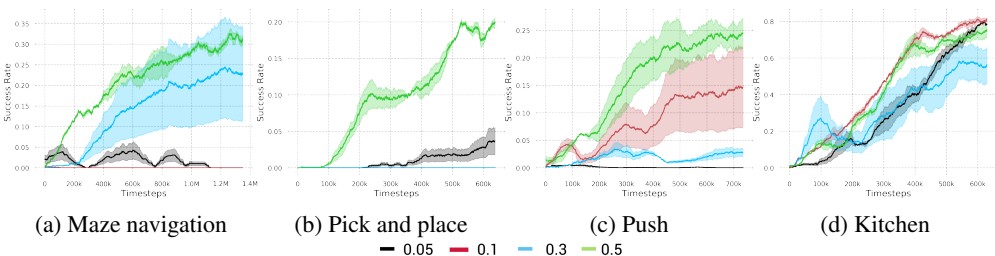

(a) Maze navigation     (b) Pick and place     (c) Push     (d) Kitchen

Figure 7: **KL weight hyper-parameter ablation** This figure compares the success rate performances for various values of KL weight $\alpha$ hyper-parameter. This hyper-parameter value controls the weight of KL constraint in higher-level policy objectives. If $\alpha$ is too large, the higher policy is very close to the reference policy, and if $\alpha$ is too small, the higher policy is far from the reference policy. We pick the hyper-parameter values after extensive ablation experiments.

Here, we provide the plots for the ablation experiments. Here, we perform the ablation analysis for selecting the hyper-parameters. The primitive regularization weight hyper-parameter $\lambda$ directly controls the magnitude of primitive regularization. If $\lambda$ is too small, we loose the advantages of primitive informed regularization. In contrast, if $\lambda$ is too large, it may lead to degenerate solutions. We provide the ablation in Figure 6. Further, the hyper-parameter $\alpha$ controls the weight of KL constraint in higher level policy objective. If $\alpha$ is too large, the higher policy is very close to the reference policy, and if $\alpha$ is too small, the higher policy might stray too far from the reference policy, leading to poor performance in both scenarios. $\alpha$ thus controls the amount of KL regularization in the maximum likelihood `DPO` objective. We provide the ablation plots in Figure 7.

## A.5 ENVIRONMENT DETAILS

This section contains additional details about the environment.

### A.5.1 MAZE NAVIGATION TASK

In this environment, a 7-DOF robotic arm gripper navigates through randomly generated four-room mazes. The gripper remains closed, and the positions of walls and gates are generated randomly. The table is discretized into a rectangular $W \times H$ grid, with vertical and horizontal wall positions $W_P$ and $H_P$ randomly selected from $(1, W-2)$ and $(1, H-2)$, respectively. In the constructed four-room environment, the four gate positions are randomly chosen from $(1, W_P - 1)$, $(W_P + 1, W - 2)$, $(1, H_P - 1)$, and $(H_P + 1, H - 2)$. The height of the gripper is fixed at table height, and it must navigate through the maze to reach the goal position, indicated by a red sphere.

The following implementation details apply to both the higher and lower-level policies unless explicitly stated otherwise. The environment features continuous state and action spaces. The state is represented as the vector $[p, \mathcal{M}]$, where $p$ is the current gripper position, and $\mathcal{M}$ is the sparse maze array. The higher-level policy input is a concatenated vector $[p, \mathcal{M}, g]$, where $g$ is the target goal

position. In contrast, the lower-level policy input is a concatenated vector $[p, \mathcal{M}, s_g]$, where $s_g$ is the sub-goal provided by the higher-level policy. The current position of the gripper is considered the current achieved goal.

The sparse maze array $\mathcal{M}$ is a discrete 2D one-hot vector array, where a value of $1$ indicates the presence of a wall block, and $0$ indicates its absence. In our experiments, the sizes of $p$ and $\mathcal{M}$ are set to 3 and 110, respectively. The higher-level policy predicts sub-goal $s_g$, so its action space dimension matches the goal space dimension of the lower primitive. The lower primitive action $a$, directly executed in the environment, is a 4-dimensional vector with each dimension $a_i \in [0, 1]$. The first three dimensions provide offsets to be scaled and added to the gripper position for movement. The last dimension controls the gripper: $0$ implies fully closed, $0.5$ implies half-closed, and $1$ implies fully open.

### A.5.2 Pick and place and Push Environments

In the pick and place environment, a 7-DOF robotic arm gripper must pick up a square block and place it at a goal position set slightly above table height. This complex task requires the gripper to navigate to the block, close the gripper to grasp the block, and then move the block to the desired goal position. In the push environment, the 7-DOF robotic arm gripper needs to push a square block towards the goal position. The state is represented as the vector $[p, o, q, e]$, where $p$ is the current gripper position, $o$ is the position of the block on the table, $q$ is the relative position of the block to the gripper, and $e$ consists of the linear and angular velocities of both the gripper and the block. The higher-level policy input is thus a concatenated vector $[p, o, q, e, g]$, where $g$ is the target goal position.

The lower-level policy input is a concatenated vector $[p, o, q, e, s_g]$, where $s_g$ is the sub-goal provided by the higher-level policy. The current position of the block is considered the current achieved goal. In our experiments, the sizes of $p$, $o$, $q$, and $e$ are set to 3, 3, 3, and 11, respectively. The higher-level policy predicts sub-goal $s_g$, so the action space and goal space dimensions are the same. The lower primitive action $a$ is a 4-dimensional vector with each dimension $a_i \in [0, 1]$. The first three dimensions provide offsets for the gripper position, and the last dimension controls the gripper (0 for closed and 1 for open). During training, the positions of the block and goal are randomly generated, with the block always starting on the table and the goal always above the table at a fixed height.

### A.6 Limitations and future work

Our DPO based hierarchical formulation raises an important question. Since DIPPER employs DPO for training the higher level policy, does it generalize on out of distribution states and actions, as compared with learning from reward model based RL formulation. A direct comparison with hierarchical RLHF based formulation might provide interesting insights. Additionally, it will be challenging to apply DIPPER in scenarios where the subgoal space is high dimensional. These are interesting research avenues, and we leave further analysis for future work.

### A.7 Impact Statement

Our proposed approach and algorithm are not expected to lead to immediate technological advancements. Instead, our primary contributions are conceptual, focusing on fundamental aspects of Hierarchical Reinforcement Learning (HRL). By introducing a preference-based methodology, we offer a novel framework that we believe has significant potential to enhance HRL research and its related fields. This conceptual foundation paves the way for future investigations and could stimulate advancements in HRL and associated areas.

### A.8 Environment visualizations

Here, we provide some visualizations of the agent successfully performing the task.

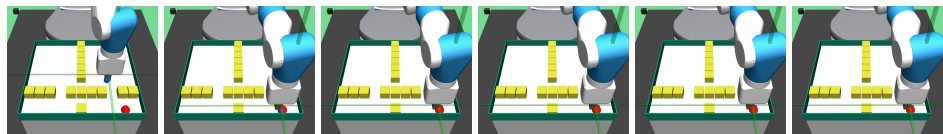

Figure 8: **Maze navigation task visualization**: The visualization is a successful attempt at performing maze navigation task

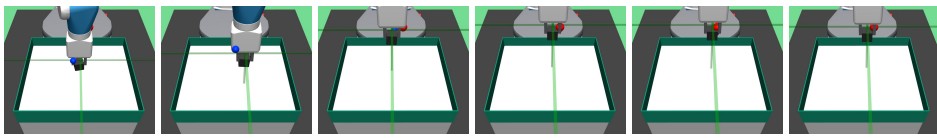

Figure 9: **Pick and place task visualization**: This figure provides visualization of a successful attempt at performing pick and place task

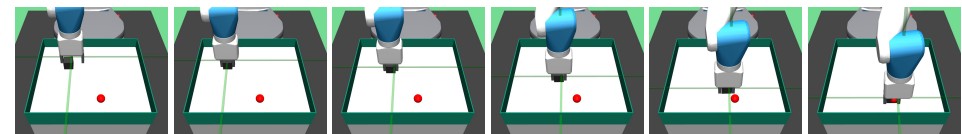

Figure 10: **Push task visualization**: The visualization is a successful attempt at performing push task

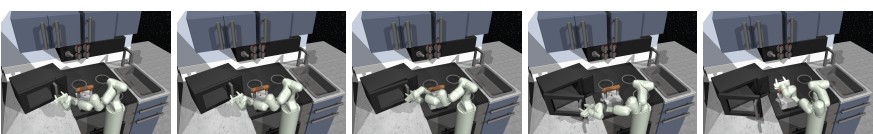

Figure 11: **Kitchen task visualization**: The visualization is a successful attempt at performing kitchen task

