# OpenReview forum: "DIPPER: Direct Preference Optimization for Primitive-Enabled Hierarchical Reinforcement Learning"
_ICLR.cc/2025/Conference — ICLR 2025 Conference Withdrawn Submission_

### Official Review · Reviewer_5TMR · 2024-11-01

**Soundness:** 3
**Presentation:** 3
**Contribution:** 1
**Rating:** 1
**Confidence:** 4

**Summary:**

The authors present DIPPER, a hierarchical RL method that directly optimizes environment reward and human preferences. The specific objective is derived from DPO, and helps mitigate non-stationarity common to most HRL methods.

**Strengths:**

**Motivation:** Non-stationarity in HRL is a big issue and this paper presents a well-motivated solution for it.

**Comparisons:** The # of and relevancy of baselines is solid, this is a convincing set of comparisons.

**Experiments:** The experiments are performed on tasks well-suited for HRL, and the analysis on goal distance prediction against HIER and HAC demonstrates that HPO’s objective encourages sampling reachable goals for the lower-level policy.

**Clarity:** THe paper is overall quite clear and the walkthrough of how to obtain the objective was both interesting and easy to read.

**Weaknesses:**

**Novelty:** I am unable to accept this paper as I just reviewed another submission for ICLR that is essentially identical except for the inclusion of actual human preferences vs using environment-given human preferences. The experiments, ablations, and **even the final objective are completely identical.** These should be one paper, ablating the use of human preferences, instead of creating 2 nearly identical papers both claiming the same novelty. I am willing to change my mind on this if sufficiently convinced, but otherwise this paper is a strong reject.

**Experiments:** Why not compare HAC and HIER on the same graphs in Figs 3/4? It’s a little strange to pick each one individually for a separate comparison when they can be compared on the same things.

**Major issues:**

- L331: Despite the subsequent claim in the paper, $\hat{m}(s_t)$ cannot be removed from the equation when optimizing for the higher level policy since it includes thye term $-\log \pi_U(g_t \mid s_t)$.
- L318-323: How the method actually **solves** with non-stationarity and infeasible subgoal is poorly explained. Please expand more on these sections with more clear explanation.

**Minor issues:**

- Line 279 uses $\pi^H$ instead of $\pi_U$
- L305: $\pi_{\pi_L}$

**Questions:**

None

**Details Of Ethics Concerns:**

Hi, I am also a reviewer for the following paper: https://openreview.net/forum?id=mJKhn7Ey4y&referrer=%5BReviewers%20Console%5D(%2Fgroup%3Fid%3DICLR.cc%2F2025%2FConference%2FReviewers%23assigned-submissions)

After reading both, I believe that the papers are so similar that they should not be two separate submissions. For example, the challenges they aim to solve in hierarchical RL are identical:

"The first (C1) is non-stationarity caused by evolving lower-level policies, which destabilizes the higher-level reward function (Chane-Sane et al., 2021). The second (C2) is the high-level policy’s tendency to generate subgoals that are infeasible for the lower-level policy to achieve."

and

"Limitation L1: non-stationarity due to evolving lower-level primitive policy, and Limitation L2: infeasible subgoal generation by higher-level policy(Chane-Sane et al., 2021). When the higher and lower level policies are trained concurrently in HRL, due to continuously changing and sub-optimal lower level policy, the higher level reward function and transition model become non-stationary."

from the first paragraph of the introduction for both.

Furthermore, their proposed solution is almost the same:

We propose a novel Hierarchical Preference Optimization (HPO) method that leverages primitiveregularized Direct Preference Optimization (DPO) to solve complex RL tasks using human preference data (Section 4). Our approach is principled; we derive it by reformulating the HRL problem as a bi-level optimization problem. To the best of our knowledge, this is the first work to utilize the bi-level optimization framework to develop a principled solution for HRL.

and

"The key idea underlying DIPPER is twofold: we introduce a DPO-based approach to directly learn higher-level policies from preferences, replacing the two-tier RLHF component in the scheme described above with a simpler, more efficient single-tier approach; we replace the reference policy inherent in DPO-based approaches, which is typically unavailable in complex robotics tasks, with a primitive-enabled reference policy derived from a novel bi-level optimization formulation of the HRL problem."

The resulting "novel" solution for the procedure proposed is in Eq 13 in this paper and Eq 15 in the other, and are exactly identical.

Finally, the experiment figures are almost identical, with near-identical performance between the two methods introduced in the two papers comparing against an identical set of baselines on an identical set of environments. This by itself isn't strange, but the near-identical performance points to how these methods are essentially the same. I believe it's the same authors too, as they cite the same paper (Singh et al 2024) as the main prior work they build upon, with the same art style for all figures.

The main differences between the two papers:

{0, 1} (this paper) reward for the low level policy vs {-1, 0} (the other paper) reward
The use of human preferences vs substituting them with environment-generated preferences.
I think this should be one paper, ablating the single choice of preferences vs environment-generated preferences. I don't believe they are sufficiently different to create two papers for.

---

### Official Review · Reviewer_dGuG · 2024-11-02

**Soundness:** 3
**Presentation:** 2
**Contribution:** 2
**Rating:** 5
**Confidence:** 3

**Summary:**

The paper provides a hierarchical (goal-conditioned) RL method for solving long horizon, spare reward tasks. It addresses an important and common issue with HRL methods, namely, the instability due to the inherent non-stationarity at the high level caused by the changing low level policies or primitives. The paper proposes to expoit preference based learning, in praticular direct preference optimization, to disentangle the high-level form the low-level learning step, thereby circumventing the high-level non-stationarity problem. The proposed method is evaluated on four sparse reward tasks and performs favorably in most of them, when compared with a number of relevant HRL and DPO baselines.

**Strengths:**

Clear and thorough derivation, good number of experiments and reasonable baselines, good theoretical contribution towards more robust and general HRL.

**Weaknesses:**

Experimental results are not very convincing to me. The main result in Fig 2 shows that Dipper (the proposed method) seems to be very slow at learning the simple maze navigation task, while baselines solve it by what seems orders an magnitude faster. In the other environments, Dipper seems to perform better than existing baselines, but never seems to solve the task fully. In the "pick and place" and "push" task it only reaches a success rate of 0.3. Granted, the baselines perform even worse, but I find that result somewhat hard to believe. Perhaps the baselines were not configured adequately? It also seems like the non of the methods robustly converged to a certain level of performance.

**Questions:**

Are there related works reporting results on some of the environments used? For example, are there other HRL methods that solve the franka kitchen environment, as far as I know that is a fairly standard method environment for benchmarking HRL.

Why is dipper so slow to learn the maze navigation problem?

The method requires eliciting human preference data. This introduces a real bottleneck. How many preference trajectory pairs does dipper need to perform well, for example on the franka kitchen environment?

I am currently rating this paper a 5, but will raise my score if the concern regarding the performance of other baselines metrics is alleviated.

---

### Official Review · Reviewer_wZcU · 2024-11-02

**Soundness:** 3
**Presentation:** 2
**Contribution:** 3
**Rating:** 5
**Confidence:** 3

**Summary:**

This paper proposes DIPPER, a novel Hierarchical Reinforcement Learning (HRL) method that can mitigate known issues in (goal-conditioned) HRL. These issues are the non-stationarity of the low-level policies preventing stable updates of the high-level policy and the proposition of invalid goals. DIPPER relies on preference optimization, more precisely on Direct Preference Optimization (DPO) which optimizes a policy based on preferences without a reward function.

**Strengths:**

- The paper proposes an interesting method that addresses important issues in Hierarchical Reinforcement Learning.
- The paper is well-written and clearly states the existing limitations and the contributions. I particularly like the way how the limitations are defined in the beginning and are recalled throughout the whole paper

**Weaknesses:**

I appreciate the efforts to provide insightful experiments, however, I believe there is more need for experiments given that the baselines outperform or perform equally well on two out of four tasks.

Some environments such as the Box pushing [1] environment could be additional and challenging environments that can provide some insights.

- Li et al. Open The Black Box: Step-Based Policy Updates For Temporally-Correlated Episodic Reinforcement Learning (ICLR 2024)

**Questions:**

- The main text mentions that the reference policy is not available for complex tasks. Is this because of the lack of the Value function, or to what exactly is this statement referred to? Technically any policy could be chosen as the reference policy.

- How could trust-region RL methods help with the presented limitations (L1 & L2)? To my understanding, the non-stationarity happens because of drastic changes in the higher-level and lower-level policies. Couldn't this be mitigated using trust region layers for reinforcement learning as presented in [2]? As a Value function is generally learned in RL, this would also mitigate Limitation 2 (infeasible goals).

Those Trust region-based updates have been proposed to stabilize learning in Hierarchical Reinforcement Learning before [3,4] and, therefore, need to be discussed in the work.

[2] Otto et al. Differentiable Trust Region Layers for Deep Reinforcement Learning (ICLR 2021)
[3] Daniel et al. Hierarchical Relative Entropy Policy Search (JMLR 2016)
[4] Celik et al. Acquiring Diverse Skills using Curriculum Reinforcement Learning with Mixture of Experts (ICML 2024)

---

### Official Review · Reviewer_undj · 2024-11-04

**Soundness:** 3
**Presentation:** 1
**Contribution:** 3
**Rating:** 6
**Confidence:** 2

**Summary:**

The authors propose a hierarchical reinforcement learning (HRL) method that addresses two key problems in the literature: the non-stationary problem of training a high-level policy while a low-level policy is also changing, and the infeasible subgoal generation problem. The proposed method addresses these two issues by training the high-level policy with direct preference optimization (DPO) and setting the reference policy to be the low-level policy.

**Strengths:**

- Using the low-level policy as a reference policy for training the high-level policy with DPO and formulating this as a bi-level optimization problem to address the non-stationary issue is a neat and novel idea. The authors present sufficient evidence that their formulation somewhat relieves the problem of infeasible goal generation (Fig. 4).

- The results demonstrate that the proposed approach performs better than baseline HRL and non-HRL methods on long-horizon sparse-reward tasks. The comparisons in Figure 2 are thorough and fair.

**Weaknesses:**

- Some aspects of the paper are poorly written. In particular, sections 4.1.1 - 4.1.3 were hard to follow. For example, it's not immediately obvious that equation 6 and the accompanying explanation is referring to equation 5 and this should be made explicit. The small paragraphs "Dealing with L1/L2" are written in a confusing way and make it sound like the solution for dealing with these problems is about to be described next, whereas in fact the authors most likely wanted to convey that their bi-level optimization formulation and solution that was previously described alleviates both the issues. Overall this section could be written better and maybe accompanied with a diagram explaining the hierarchical policy and bi-level optimization problem formulation as a visual aid.

- Figure 4 only compares subgoal distances to HIER, even though the authors include other HRL baselines in other experiments. Why weren't those comparisons made here as well? This seems a rather weak experiment to me.

**Questions:**

Question was asked above in the weaknesses section.

---

### Note · Authors · 2024-11-18

I have read and agree with the venue's withdrawal policy on behalf of myself and my co-authors.